# Research on Driving Fatigue Alleviation Using Interesting Auditory Stimulation Based on VMD-MMSE

**DOI:** 10.3390/e23091209

**Published:** 2021-09-14

**Authors:** Fuwang Wang, Bin Lu, Xiaogang Kang, Rongrong Fu

**Affiliations:** 1School of Mechanic Engineering, Northeast Electric Power University, Jilin 132012, China; 2201900513@neepu.edu.cn (B.L.); 2201900509@neepu.edu.cn (X.K.); 2College of Electrical Engineering, Yanshan University, Qinhuangdao 066004, China; frr1102@ysu.edu.cn

**Keywords:** driving fatigue alleviation, MMSE, VMD, LSM, SQ, interesting auditory stimulation

## Abstract

The accurate detection and alleviation of driving fatigue are of great significance to traffic safety. In this study, we tried to apply the modified multi-scale entropy (MMSE) approach, based on variational mode decomposition (VMD), to driving fatigue detection. Firstly, the VMD was used to decompose EEG into multiple intrinsic mode functions (IMFs), then the best IMFs and scale factors were selected using the least square method (LSM). Finally, the MMSE features were extracted. Compared with the traditional sample entropy (SampEn), the VMD-MMSE method can identify the characteristics of driving fatigue more effectively. The VMD-MMSE characteristics combined with a subjective questionnaire (SQ) were used to analyze the change trends of driving fatigue under two driving modes: normal driving mode and interesting auditory stimulation mode. The results show that the interesting auditory stimulation method adopted in this paper can effectively relieve driving fatigue. In addition, the interesting auditory stimulation method, which simply involves playing interesting auditory information on the vehicle-mounted player, can effectively relieve driving fatigue. Compared with traditional driving fatigue-relieving methods, such as sleeping and drinking coffee, this interesting auditory stimulation method can relieve fatigue in real-time when the driver is driving normally.

## 1. Introduction

Driving fatigue is a physiological phenomenon in which the driver’s attention and alertness decrease after a long driving task. When humans are in a state of driving fatigue, the speed of the brain’s processing of information, especially its ability to deal with emergencies, decreases obviously. Studies have shown that driving fatigue is one of the main causes of traffic accidents [1,2,3]. A study by Byeon showed that driving fatigue causes at least 10%–15% of traffic accidents [4]. Therefore, it is of great significance for traffic safety to accurately detect drivers’ fatigue states and to take certain measures to relieve fatigue when necessary.

Driving fatigue detection can be divided into subjective methods and objective methods [5,6]. The subjective method is generally conducted in the form of questionnaires, which are often used as an auxiliary method in the detection of driving fatigue [7]. Objective detection methods mainly detect the driver’s mental state by detecting the driver’s physiological signals [8], facial expressions [9], vehicle tracks [10] and so on. Considering the accuracy and reliability of the objective detection method of driving fatigue, more and more researchers have applied this method to their own research [11,12]. In recent years, there have been more and more studies on fatigued driving using drivers’ physiological signals, such as EEG [13,14], EOG [15] EMG [16] and ECG methods [17,18]. Fatigue detection method based on EEG characteristics is recognized as the gold standard by researchers [19]. Li et al. used EEG data on human mental fatigue to construct a brain function network, and proposed a modified greedy coloring algorithm to calculate the fractal dimension of binary and weighted brain function networks. The results showed that the fractal dimension increased with the increase of mental fatigue [20]. Mental fatigue characteristics in some brain leads are not obvious, and the noise interference of redundant channels increases the computational complexity. Li et al. identified the best indicator of driving fatigue through gray relational analysis, and reduced the dimensions of effective electrodes through kernel principal component analysis, and finally obtained two important leads (FP1 and O1) [21]. Jap et al. calculated the coherence between the inter-hemispheres of θ, δ, α and β sub-bands of five homologous EEG electrode pairs (FP1–FP2, C3–C4, T7–T8, P7–P8 and O1–O2). The results showed that the inter-hemispheric coherence values of the frontal and occipital areas were higher than that of the central, parietal and temporal areas in the four frequency bands [22]. Wang et al. collected EEG signals from the central, parietal and occipital regions of the subjects, and used the G-P algorithm to evaluate the correlation dimension to quantify EEG signals. The results showed that the correlation dimension of each channel decreased significantly with an increase in the subjects’ driving fatigue [23]. In this paper, we use O1 and O2 electrodes in the occipital region to study driving fatigue.

In recent years, non-stationary signal analysis has received more and more attention from researchers, in which amplitude-modulated-frequency-modulated (AM-FM) signal analysis is of great significance in non-stationary signal analysis [24,25]. The frequency and amplitude information provided by non-stationary signals can be obtained via demodulate FM-AM signals. Qin achieved FM-AM decomposition of multi-component signals by combining energy decomposition and adaptive filtering [26]. At present, AM-FM decomposition techniques mainly include the iterated Hilbert transform (IHT) [27], the empirical mode of decomposition (EMD) [28], the local mean decomposition (LMD) approach [29], etc. The IHT primarily separates the amplitude signal from the original signal. Then, a filter is used to separate the single component from the amplitude signal. Multiple FM-FM single components can be extracted from the original signal through the iterative use of the Hilbert transform and filter. EMD is an adaptive signal processing method that adaptively decomposes complex multi-component signals into the sum of several IMFs. Then, the Hilbert transformation is performed to calculate the instantaneous frequency and the instantaneous amplitude of each IMF component. EEG signals are usually mixed with certain components of noise, and traditional digital filtering or wavelet denoising methods cannot effectively reduce noise. Therefore, many scholars tend to use adaptive methods to process EEG signals in order to improve the signal-to-noise ratios of EEG signals [30]. In this study, the EEG signals were adaptively decomposed using the VMD approach to obtain multiple IMF components, in which each IMF was defined as an FM-AM signal. Compared with the empirical mode decomposition (EMD) approach, the VMD method is a completely non-recursive time domain signal processing method, which can effectively avoid the occurrence of modal aliasing [31]. Kaur et al. compared discrete wavelet transform (DWT), combined with VMD and wavelet packet transform (WPT) to denoise physiological signals, and achieved ideal denoising effects [32]. Therefore, we have use VMD to adaptively decompose EEG signals to achieve signal noise reductions.

EEG signals are highly complex signals and are non-linear and non-stationary; therefore, linear analysis methods cannot adequately reflect the internality dynamic characteristics of EEG signals. As one of the non-linear analysis methods, th mutlifractality method can reveal the regularity problems within the non-linear system by studying its dynamic characteristics, such as its disorder, irregularity and uncertainty. At present, the mutlifractality method is widely used in physiological signal processing [33,34,35]. Additionally, as a non-linear processing method, entropy is also widely used in the detection of driving fatigue [36,37,38]. Compared with the mutlifractality method [39,40], MMSE can extract EEG features on a time scale, and the data length required by the MMSE method is shorter in terms of EEG processing. Consequently, for the special situation of driving fatigue detection, the MMSE method can better meet the needs of traffic safety. Wang et al. analyzed drivers’ driving levels using the sample entropy value of the δ sub-band in the drivers’ EEG signals. The results showed that the sample entropy value can distinguish between different driving levels of drivers [41]. Gao et al. proposed a new relative wavelet entropy complex network method to identify driving fatigue, and the results showed that this method could effectively improve the classification accuracy for driving fatigue [42]. The traditional single-scale entropy cannot adequately reflect the fatigue characteristics in EEG signals. Therefore, in recent years, more and more scholars have used the multi-scale entropy method to extract EEG features [43]. Zou et al. proposed a multi-scale entropy-based empirical mode decomposition (EMD) method to identify the characteristics of driving fatigue, and the results show that this method can effectively detect driving fatigue [44]. Due to the coarse-graining procedure, the signal length of the traditional multi-scale entropy gradually becomes shorter. When applied to a short-term time series, the entropy estimation becomes inaccurate [45]. Wu et al. proposed an MMSE method to transform the coarse-graining procedure into a moving-average process and introduced a time delay to construct template vectors in the calculation of sample entropy [46]. In our study, we use the moving-average method to multi-scale EEG signals in order to overcome the disadvantage that the traditional multi-scale entropy causes the time series length to become shorter with the increase of the scale factor.

Driving fatigue poses a great threat to road traffic, and relieving driving fatigue effectively is of great significance for traffic safety. The traditional methods of relieving driving fatigue include stopping to rest, ventilating the cab, as well as drinking coffee or other irritating drinks. These methods require the driver to stop driving and the relief effect is often not ideal; therefore, they are not suitable for long-distance driving environments. In recent years, many researchers have begun to relieve driving fatigue using electrical stimulation of human acupoints, such as stimulating the Hégǔ point (L14), the Neìguān point (PC6) and the Fēngchí point (GB20) [47,48,49]. However, frequent electrical stimulation of human acupoints to relieve fatigue will gradually reduce the relief effect. Studies have shown that appropriately increasing drivers’ cognitive load—that is, performing a series of secondary tasks while driving—can improve drivers’ alertness [50,51]. Oron–Gilad’s research shows that the fatigue relief effect is the best when the driver’s workload is in the optimal state [52]. In this study, we tried to stimulate drivers with interesting auditory stimulation to achieve the effect of alleviating driving fatigue.

In this study, the VMD algorithm was used to adaptively decompose drivers’ EEG signals, obtained multiple IMFs. Then the multi-scale entropy features of each IMF were calculated using a multi-scale sample entropy algorithm. Finally, multi-scale sample entropy was used to detect driving fatigue. The alleviation of driving fatigue is of great significance to traffic safety. Therefore, driving fatigue alleviation was studied from the perspective of driver cognitive load in this study, that is, using the method of interesting auditory stimulation to bring the driver’s cognitive level to a relatively optimal state to achieve the alleviation of driving fatigue.

## 2. Materials and Methods

### 2.1. Experiment

#### 2.1.1. Subjects

A total of 15 subjects (10 males and 5 females; aged 30 ± 3.6 (standard deviation, SD)) participated in the experiment. They were required to perform driving tasks on a driving simulator during the experiment. To trigger fatigue for the subjects, a monotonous highway scene was selected for the experiment. All subjects had no history of sleep-related disease or mental illness and had not consumed any irritant drinks or any irritant foods or drugs within 48 h prior to the experiment. All subjects were informed regarding the purpose of this experiment and written consent forms were signed. The experiment was divided into two experimental modes—the normal driving mode and the interesting auditory stimulation mode. In the former experimental mode, the subjects performed a normal monotonous driving task. In the latter experimental mode, the subjects performed normal driving tasks in the driving simulator while the experimenter played Chinese traditional cross-talk for them. All subjects were required to complete two types of driving experiments. It should be noted that the interesting auditory stimulation mode of the experiment could only be carried out after all subjects have completed the normal driving mode of experiment. Each of the modes required 3 h of driving. To render the subjects more susceptible to fatigue, the experiment was selected during the time period from 1:00 p.m. to 4:00 p.m., and was divided into seven stages of collection, each of which took 5 min (stage 1—1:00 p.m.; stage 2—1:30 p.m.; stage 3—2:00 p.m.; stage 4—2:30 p.m.; stage 5—3:00 p.m.; stage 6—3:30 p.m.; stage 7—4:00 p.m.). The subjects were asked about their current mental status in the five minutes prior to each EEG acquisition and were scored based on the Karolinska sleepiness scale (KSS). To avoid the potential influence of a lack of sleep on the results of the experiment, the subjects were asked to rest for half an hour before the the experiment (12:00 p.m.–12:30 p.m.).

#### 2.1.2. EEG Acquisition Device

The Emotiv device was used to collect EEG data in this experiment. The sampling rate of the device was 128 Hz and the 14 electrodes were arranged according to the international 10–20 system (14 channels = AF3, AF4, F3, F4, FC5, FC6, F7, F8, T7, T8, P7, P8, O1, O2). The electrode caps were correctly attached to the subjects before the experiments were conducted, and the connection status of the 14 electrodes was checked at all times during the experiments to ensure the validity of the data. In addition, we ensured that the experiment environment was quiet to reduce noise disturbances. The equipment used for this experiment is shown in Figure 1.

### 2.2. Methods

The EEG is susceptible to interference from external environmental noise, and there are phenomena such as baseline drift in the raw EEG, which can seriously influence the extraction of fatigue features. Thus, pre-processing of the EEG is crucial. In this paper, EEGLAB software was used to perform the baseline correction of the raw EEG signals, followed by independent component analysis (ICA) to remove eye movements, blinks and EMG artifacts. Studies have shown that fatigue features are easily found in the 0.5–44 Hz sub-band of EEG signals [53,54]; therefore, low-pass filtering was performed on the EEG to obtain 0.5–44 Hz EEGs. In this study, the VMD algorithm was used to adaptively decompose the pre-processed EEG signals to obtain several IMF components. As each IMF component contains different fatigue characteristics, it is necessary to select the IMF component. First, a single-scale sample entropy algorithm was used to calculate the single-scale entropy characteristics of each IMF component. Secondly, the LSM was used to fit the change trends of the characteristics of the subjects’ 7 driving stages. Finally, the IMF component with the most obvious change trends in terms of fatigue characteristics was selected according to the slope of the fitting. Multi-scale entropy characteristics were calculated according to the selected IMF component. Then, the LSM was used to fit the change trends of multi-scale entropy characteristics with the 7 driving stages of the subjects. According to the fitting slope, the scale factor of multi-scale entropy features with the most obvious change trend was selected.

#### 2.2.1. Variational Mode Decomposition

The 0.5–44 Hz sub-band EEG signals contain a certain component of low-frequency noise, which affects the extraction of fatigue features. Thus, the 0.5–44 Hz narrow-band signal was decomposed using VMD in this study. The main idea of VMD is to decompose the signal into multiple narrow-band IMFs based on the central frequency, which is continuously updated during the decomposition process. The VMD adaptively obtains the variational mode function by solving the optimal solution of the constrained variational model. The calculation process was as follows.

The VMD defines each mode as an AM-FM signal.
(1)$$u_{k}=A_{k}(t)\cos(\phi_{k}(t))$$
where, $A_{k}(t)$ is the instantaneous amplitude, $d\phi_{k}(t)/dt=\omega_{k}(t)$, and $\omega_{k}(t)$ is the instantaneous frequency. The VMD constraint is that the sum of the modal components is equal to the input signal and that the sum of the estimated bandwidths of each mode is minimized. The model with these constraits is shown as follows.
(2)$$\begin{array}{l} \underset{\left\{u_{k}\}\{\omega_{k}\right\}}{\min}\left\{{\sum_{k}\left\|\partial_{t}[(\delta (t)+\frac{j}{\pi t})*u_{k}(t)]e^{-j\omega_{k}t}\right\|}_{2}^{2}\right\}\\ s.t.\sum_{k}u_{k}=f \end{array}$$
where $u_{k}$ represents the *K* IMF from the decomposition, and $\omega_{k}$ represents the central frequency of the corresponding IMF.

The Lagrangian multipliers λ(t) and the penalty factor α are introduced to transform the constrained variational problem into an unconstrained variational problem by calculating the optimal solution in Equation (2), the expression of which is shown as
(3)$$\begin{array}{l} L(\{u_{k}\},\{\omega_{k}\},\lambda )=\alpha {\sum_{k}\left\|\partial_{t}[(\delta (t)+\frac{j}{\pi t})*u_{k}(t)]e^{-j\omega_{k}t}\right\|}_{2}^{2}\\ +{\left\|f(t)-\sum_{k}u_{k}(t)\right\|}_{2}^{2}+\langle \lambda (t),f(t)-\sum_{k}u_{k}(t)\rangle \end{array}$$
where $\langle •\rangle$ denotes the inner product operation, and the alternate direction method of multipliers (ADMM) iteratively updates $u_{k}^{n+1}$, $\omega_{k}^{n+1}$ and $\lambda^{n+1}$ to search for the augmented Lagrangian function, that is, the “saddle point” of Equation (3), and the expression of $u_{k}^{n+1}$ is,
(4)$$u_{k}^{n+1}=\underset{u_{k}\in X}{\arg\min}\left\{\alpha {\left\|\partial_{t}[(\delta (t)+\frac{j}{\pi t})*u_{k}(t)]e^{-j\omega_{k}t}\right\|}_{2}^{2}+{\left\|f(t)-\sum_{i}u_{i}(t)+\frac{\lambda (t)}{2}\right\|}_{2}^{2}\right\}$$
where $\omega_{k}^{}=\omega_{k}^{n+1}$, $\sum_{i}u_{i}(t)=\sum_{i\neq k}u_{i}{\left(t\right)}^{n+1}$. According to Parseval’s theorem and the Plancherel theorem, Equation (4) can be converted to the frequency domain.
(5)$$\begin{array}{l} {\hat{u}}_{k}^{n+1}=\underset{{\hat{u}}_{k}}{\arg\min}\{\alpha {\left\|j(\omega -\omega_{k})[(1+\mathrm{sgn}(\omega )){\hat{u}}_{k}(\omega )]\right\|}_{2}^{2}\\ \left.+{\left\|\hat{f}(\omega )-\sum_{i}{\hat{u}}_{i}(\omega )+\frac{\hat{\lambda }(\omega )}{2}\right\|}_{2}^{2}\right\} \end{array}$$

Using the Hermitian symmetry of real-valued signals, Equation (5) is rewritten into a half-space integral over non-negative frequencies.
(6)$$\begin{array}{l} {\hat{u}}_{k}^{n+1}=\underset{{\hat{u}}_{k}}{\arg\min}\left\{\int_{0}^{+\infty }4\alpha {\left(\omega -\omega_{k}\right)}^{2}{\left|{\hat{u}}_{k}(\omega )\right|}^{2}\right.\\ \left.+2{\left|\hat{f}(\omega )-\sum_{i}{\hat{u}}_{i}(\omega )+\frac{\hat{\lambda }(\omega )}{2}\right|}^{2}d\omega \right\} \end{array}$$

Finally, we obtain ${\hat{u}}_{k}^{n+1}$ as follows.
(7)$${\hat{u}}_{k}^{n+1}(\omega )=\frac{\hat{f}(\omega )-\sum_{i\neq k}{\hat{u}}_{i}(\omega )+\frac{\hat{\lambda }(\omega )}{2}}{1+2\alpha {\left(\omega -\omega_{k}\right)}^{2}}$$

The iterative optimization formula for $\omega_{k}$ is as follows.
(8)$$\omega_{k}^{n+1}=\underset{\omega_{k}}{\arg\min}\left\{{\left\|\partial_{t}\left[\left(\delta (t)+\frac{j}{\pi t}\right)*u_{k}(t)\right]e^{-j\omega_{k}t}\right\|}_{2}^{2}\right\}$$

Transform the optimization problem of Equation (8) into the Fourier domain to obtain:(9)$$\omega_{k}^{n+1}=\frac{\int_{0}^{\infty }\omega {\left|{\hat{u}}_{k}(\omega )\right|}^{2}d\omega }{\int_{0}^{\infty }{\left|{\hat{u}}_{k}(\omega )\right|}^{2}d\omega }$$
(10)$$\mathrm{Update}\;{\hat{u}}_{k}^{n+1},\;\omega_{k}^{n+1}\mathrm{and}\;{\hat{\lambda }}^{n+1}(\omega )\;\mathrm{using}\;\mathrm{the}\;\mathrm{following}\;\mathrm{equation},\;\mathrm{respectively}.\;\;{\hat{u}}_{k}^{n+1}(\omega )=\frac{\hat{f}(\omega )-\sum_{i<k}{\hat{u}}_{i}^{n+1}(\omega )-\sum_{i>k}{\hat{u}}_{i}^{n}(\omega )+\frac{{\hat{\lambda }}^{n}(\omega )}{2}}{1+2\alpha {\left(\omega -\omega_{k}^{n}\right)}^{2}}$$
(11)$$\omega_{k}^{n+1}=\frac{\int_{0}^{\infty }\omega {\left|{\hat{u}}_{k}^{n+1}(\omega )\right|}^{2}d\omega }{\int_{0}^{\infty }{\left|{\hat{u}}_{k}^{n+1}(\omega )\right|}^{2}d\omega }$$
(12)$${\hat{\lambda }}^{n+1}(\omega )\leftarrow {\hat{\lambda }}^{n}(\omega )+\tau (\hat{f}(\omega )-\sum_{k}{\hat{u}}_{k}^{n+1}(\omega ))$$

Based on the above analysis, it can be shown that in the process of solving the variational model, the central frequency of each IMF component, as well as the bandwidth, is continuously iteratively updated until the condition ∑ku^kn+1−u^kn22u^kn22<e is satisfied and the iteration is stopped, and for a given accuracy e>0, the loop is finished. The real signal is decomposed into K IMF components according to the frequency domain characteristics, completing the adaptive decomposition of the signal and effectively avoiding the phenomenon of modal aliasing.

#### 2.2.2. Sample Entropy

The SampEn, which is modified from ApEn, reflects the complexity of the time series. The SampEn method was first proposed by Richman and Moorman to analyze random data sets with known probabilities. The research of Richman and Moorman also showed that a larger SampEn value indicates a more irregular time series, whereas a smaller the entropy value indicates the stronger regularity of the time series [55]. In recent years, the SampEn has been widely used in the field of EEG signal processing. Shalbaf et al. used SampEn characteristics to reflect the effect of sevoflurane anesthesia. The results showed that the SampEn had a faster reaction to the transients of EEGs during the induction of anesthesia [56]. In this study, the SampEn algorithm was used to extract the EEG fatigue characteristics. The SampEn algorithm is calculated as shown in the following procedure.

A time-series of length *N* is denoted x(1),x(2),…,x(N), and the embedding dimension is set to *m* and the time delay to *τ*.
(13)$$X_{m}^{\tau }(i)=\{x(i),x(i+\tau ),\ldots ,x[i+(m-1)\tau ]\};1\leq i\leq N-(m-1)\tau$$

Calculating the distance between any pair of m-dimension vectors:(14)$$\begin{array}{l} d[X_{m}^{\tau }(i),X_{m}^{\tau }(j)]=\max[x(i+k\tau )-x(j+k\tau )];\\ 0\leq k\leq m-1;i\neq j;1\leq i,j\leq N-(m-1)\tau \end{array}$$

Given a threshold *r*, we calculate the number at which the distance maximum differences between the above two *m*-dimension vectors is less than *r*, and calculate the ratio of this number to the total numbers.
(15)$$B_{i}^{m}(r)=\frac{1}{N-(m-1)\tau }\{number\;of\;d[X_{m}^{\tau }(i),X_{m}^{\tau }(j)]<r,\;i\neq j\}$$

$B_{i}^{m}(r)$ is the ratio of the number, less than the threshold *r*, to the total number, with the mean value calculated as follows.
(16)$$B^{m}(r)=\frac{1}{N-(m-1)\tau }\sum_{i=1}^{N-m+1}B_{i}^{m}(r)$$
in which $B^{m}(r)$ is the mean of the *m*-dimension sequence ratio.

The signal is added to *m* + 1 dimensions and the above steps are repeated to obtain the mean value $B^{m+1}(r)$ of the ratio of the *m* + 1 dimension sequence.

The SampEn of the EEG signal is
(17)$$SampEn(X,m,r,\tau )=-\ln\frac{B^{m+1}(r)}{B^{m}(r)}$$

#### 2.2.3. Modified Multi-Scale Entropy

The modified multi-scale entropy (MMSE) approach was proposed by Wu et al. [46], and the calculation procedure requires two steps: (1) a dynamic system representing different time scales is obtained through the moving-average procedure; (2) the entropy value of a moving-average time series with a scale factor *τ* is calculated using the SampEn method with a time delay *τ*.

Using $X^{\tau }$ to represent a moving-average time series with a time scale of *τ*, the formula is shown as follows.
(18)$$x_{j}^{\tau }=\frac{1}{\tau }\sum_{i=j}^{j+\tau -1}x_{i}\;\;\;\;\;\;(1\leq j\leq N-\tau +1)$$

The entropy value of the moving-average time series $X^{\tau }$ is calculated using the SampEn with a time delay *τ*, calculated as,
(19)$$SampEn(X^{\tau },m,r,\tau )=-\ln\frac{B^{m+1}(r)}{B^{m}(r)}$$

According to a previous study [57], the similarity tolerance *r* = 0.2*SD* (*SD* is the standard deviation of the raw EEG) and the embedding dimension *m* = 2 were chosen for this study.

#### 2.2.4. Least Squares Method

The main idea of the LSM is to fit the existing data points to obtain the correspondence between variables and other variables. The data can be predicted according to the correspondence between different variables. In this paper, the LSM was used to fit the fatigue characteristics in seven driving stages to obtain the slope of the change trends of the fatigue characteristics.

#### 2.2.5. Statistical Analysis Algorithm

In this study, the statistical method of two-tailed *t*-tests was used to compare and analyze the differences in the experimental results. In this part of the comparative analysis, the two-tailed *t*-test was used to compare and analyze the significant differences between the normal driving mode and the interesting auditory stimulation mode.

## 3. Results

### 3.1. Selection of Intrinsic Mode Function Components

The VMD is a completely non-recursive time domain signal processing method. Figure 2 shows a time domain signal diagram, decomposed using the VMD from the first stage of the O1 channel, with the subject driving in normal driving mode.

Each IMF component contains different fatigue components; therefore, it is necessary to select the IMF component with the most obvious driving fatigue characteristics from the five IMF components. In this study, we used the LSM to perform one linear fitting process to the single-scale entropy change trends of the five IMFs in the seven driving stages, and then calculated the slope of the fatigue change trends in the seven driving stages. The means of the absolute values of the calculated slopes for 15 subjects are shown in Figure 3.

As can be seen from Figure 3, all five IMF components contain a certain component of fatigue characteristics. However, the change trend of the fatigue characteristics contained in IMF3 was more obvious and more conducive to driving fatigue detection, compared to the others. Compared to the other IMF components, the standard deviation of IMF3 was smaller, which indicated that IMF3 was less affected by individual differences and contained a more stable trend in terms of fatigue characteristics. In order to better detect and analyze the characteristics of driving fatigue, IMF3 was used to detect driving fatigue characteristics in this paper.

### 3.2. Multi-Scale Selection

To analysis the influence of scale factors on EEG extraction in relation to driving fatigue features, we calculated the change trend of MMSE for IMF3 at scale factors 1–7 over seven driving stages. The fatigue feature change trends of the seven driving stages based on MMSE on a 1–7 scale were fitted using the LSM and then the obtained slopes were calculated in terms of absolute values. The distribution of the slopes’ absolute values for the 15 subjects under the seven scale factors is shown in Figure 4.

As shown in Figure 4, the absolute slope, corresponding to the change trend of the MMSE, increases continuously with the increase of the scale factor when the scale factor is selected in the range of 1–4. The absolute slope is largest when the scale factor is four. After the scale factor becomes greater than four, the absolute slope starts to decrease gradually. This result indicates that the change trend of the MMSE is more obvious for the 15 subjects when the scale factor is four. The distributed of data for a scale factor of 4 is more centralized than the distributed of data for other scales, which indicates that the fatigue features extracted using MMSE are less affected by individual differences when the scale factor is four, which is more conducive to the detection of driving fatigue. Therefore, we selected a scale factor of four for fatigue feature extraction.

### 3.3. Modified Multi-Scale Entropy Feature

After the above analysis, IMF3 was selected to extract MMSE features with a scale factor of four. Studies have shown that more obvious fatigue features can be extracted from EEGs in the occipital lobe [58,59]. Therefore, the O1- and O2-channel EEG signals were selected to extract MMSE features for the two driving modes, respectively. The results are shown in Figure 5.

As can be seen in Figure 5, the MMSE of the O1 and O2 channels displayed significant downward trends with the extension of the driving time in normal driving mode, indicating a gradually deepening of driver fatigue. When the driver performs monotonous driving tasks for a long time in normal driving mode, the driver experiences an insufficient cognitive load, which easily leads to passive fatigue. With the extension of driving time, the driver’s reaction time in response to emergency situations becomes gradually longer, they become less alert, their perception of the outside world is weakened and their fatigue level gradually increases. The MMSE showed a slight downward trend with the extension of the driving time during interesting auditory stimulation, indicating that driver fatigue increased more slowly when driving tasks were performed. When exposed to interesting auditory stimulation, drivers were at their cognitive optimum and were not prone to passive fatigue. However, with the extension of driving time, drivers emerged slightly fatigued. As a result, the MMSE demonstrated a slight decrease during interesting auditory stimulation. Additionally, there was a significant difference between the two driving modes (O1: |*t*| = 5.138 > *t*_0.05,15_ = 2.131, *p* =1.64 × 10^−5^ < 0.05; O2: |*t*| = 6.47 > *t*_0.05,15_ = 2.131, *p* = 1.731 × 10^−6^ < 0.05). We also used one-way ANOVA to analyze the fatigue characteristics calculated based on MMSE for the two experiments. The results showed a significant difference between the two driving modes (O1: *F* = 4.78 > *F*(1,12) = 4.75, *p* = 0.0493 < 0.05; O2: *F* = 4.98 > *F*(1,12) = 4.75, *p* = 0.0455 < 0.05). Consequently, the interesting auditory stimulation method used in this paper can effectively relieve driving fatigue.

### 3.4. Subjective Questionnaire

This experiment evaluated the different fatigue states of drivers using a 9-level KSS (1—extremely alert; 2—very alert; 3—alert; 4—rather alert; 5—neither alert nor sleepy; 6—some signs of sleepiness; 7—sleepy, no effort to stay awake; 8—sleepy, some effort to stay awake; 9—very sleepy, great effort to stay awake). Figure 6 shows the changing trends of the questionnaire scores for the 15 subjects in the seven driving stages.

As can be seen in Figure 6, the questionnaire scores showed an upward trend when subjects were performing the two driving modes, indicating that the fatigue level gradually increased over the extension of he driving time. However, there was a significant difference between the two driving modes (|*t*| = 5.897 > *t*_0.05,15_ = 2.131, *p* = 2.41 × 10^−6^ < 0.05). In addition, we conducted one-way ANOVA on the KSS scores for the two types of experiments. The results showed a significant difference between the two types of experiments (*F* = 4.87 > *F*(1,12) = 4.75, *p* = 0.0475 < 0.05). Due to the influence of interesting auditory stimulation, drivers’ cognitive levels reached relatively optimum levels. Accordingly, drivers’ fatigue levels increased relatively slowly during the interesting auditory stimulation mode, compared to the normal driving mode, which indicated that the interesting auditory stimulation method used in this paper was effective for alleviating driving fatigue.

## 4. Discussion

Previous studies have shown that when people are experiencing driving fatigue, their actions become dulled, their concentration decreases and they are more likely to make misjudgments, which poses a threat to traffic safety [60]. Every year, traffic accidents caused by driving fatigue account for a large proportion of the total number of traffic accidents [61]. Therefore, it is necessary to detect the mental fatigue of drivers and alleviate it in time. Research shows that driving fatigue easily occurs when drivers have a low cognitive load [62]. In our study, we tried to appropriately increase subjects’ cognitive loads to combat driving fatigue caused by a monotonous driving environment.

### 4.1. VMD-MMSE Method

SampEn is suitable for feature detection in small sample datasets and has little dependence on data length, which is suitable for real-time online detection [40]. Studies have shown that the SampEn method is able to extract fatigue features from EEGs [63,64]. Wang et al. used SampEn to calculate the characteristics of heart rate variability (HRV) to analyze the changes in the levels of driving fatigue. The results showed that SampEn could effectively detect subjects’ driving fatigue in real time [65]. Wang et al. calculated the mean SampEn values of two EEG channels to detect drivers’ mental fatigue states. The results showed that the method was effective in detecting driving fatigue [66]. The traditional SampEn method can only be used to calculate entropy values based on a singe scale, neglecting the influence of the time scale on the results. On the other hand, multi-scale entropy analysis takes into account the influence of the time scale on fatigue characteristics. In this study, the VMD-MMSE method was used to extract fatigue features. Studies have shown that three factors—drivers’ age, gender and driving time—are the main factors affecting driving fatigue [67,68]. In this study, the effects of drivers’ age, gender and driving time on driving fatigue were analyzed using a linear-mixed effects model. The results of our analysis using SPSS software are shown in Table 1.

As can be seen in Table 1, the drivers’ age and gender had no significant influence on the degree of driving fatigue (*p* > 0.05). However, the driving time had a significant influence on the degree of driving fatigue (*p* < 0.05). Therefore, the LSM was used to linearly fit the driving time and entropy characteristics of subjects in this study. In order to verify the effectiveness of the VMD-MMSE method, based on the O1 channel EEG data in normal driving mode, the change trends of the fatigue characteristics, calculated using SampEn and the VMD-MMSE method, were fitted with the seven driving stages. The results are shown in Figure 7.

The change trends of fatigue features, extracted using the VMD-MMSE method, were more obvious, as can be seen in Figure 7. In this paper, the mean and standard deviation of the slopes of the changes in fatigue characteristics across the seven driving stages for 15 subjects were calculated using the SampEn and VMD-MMSE methods. The results are shown in Table 2.

It can be seen in Table 2 that the average of the changes in fatigue characteristics of the 15 subjects extracted using VMD-MMSE method was smaller than that extracted using the SampEn method. The results show that the fatigue characteristics extracted using VMD-MMSE method were more obvious. Moreover, the standard deviation of the fatigue change trends calculated using the VMD-MMSE method was smaller, which indicates that fatigue characteristics calculated using this method were less affected by individual differences. As EEG signals do not show complex dynamics with perfect regularity and complete randomness, complex dynamics usually reveals structures on multiple spatial and temporal scales. The traditional SampEn method only calculates the EEG characteristics on a single scale, ignoring the inherent complex time fluctuation of the EEG. However, MMSE can provide more details about complex dynamics, which is more suitable for quantifying EEGs with multiple temporal and spatial scales. Consequently, these multi-scale features, which are ignored in the SampEn method, are explicitly addressed in the MMSE algorithm.

### 4.2. Interesting Auditory Stimulation Alleviates Driving Fatigue

The results of the VMD-MMSE analysis showed that there was a significant difference in the fatigue change trends between the normal driving mode and the interesting auditory stimulation mode (O1: |*t*| = 5.138 > *t*_0.05,15_ = 2.131, *p* = 1.64 × 10^−5^ < 0.05; O2: |*t*| = 6.47 > *t*_0.05,15_ = 2.131, *p* = 1.731 × 10^−6^ < 0.05). Previous studies have shown that entropy values show a downward trend with the increase of the driving fatigue level, which is consistent with our research results [69]. As can be seen in Figure 5, when performing normal driving tasks, the fatigue features extracted using the VMD-MMSE method showed an obvious downward trend after subjects performed the monotonous driving task for a long time, due to inadequate cognitive stimulation of the driver, indicating that the fatigue level of the driver increased with the extension of the driving time. Compared with normal driving mode, there was no passive fatigue during the interesting auditory stimulation mode. When stimulated by minor tasks, the central nervous system of the driver is more active and the cognitive level of the driver reaches a relatively optimum level. Consequently, the fatigue features extracted based on the VMD-MMSE method showed a slower downward trend, indicating that the driving fatigue was relieved. Furthermore, as can be seen in Figure 6, the variation trends of the questionnaire scores in the two driving modes corresponding to the variation trends of the fatigue characteristics extracted using the VMD-MMSE method. The questionnaire scores of the two driving modes presented significant differences (|*t*| = 5.897 > *t*_0.05,15_ = 2.131, *p* = 2.41 × 10^−6^ < 0.05). Therefore, the interesting auditory stimulation method used in this paper can effectively relieve driving fatigue. Additionally, the equipment used in this experiment is portable, easy to operate and convenient to implement in real-time, which is of great significance for practical applications in the future.

### 4.3. Limitations

Although interesting auditory stimulation was proven to be effective in relieving long-term driving fatigue, we have not performed much research into the semantic content of auditory stimulation. Whether semantic content that is more exciting, neutral or boring is more conducive to relieving mental fatigue has not yet been studied.

### 4.4. Future Lines of Research

In future research, our work will be divided mainly into three aspects. One involves choosing different auditory material in terms of semantic content to determine the relationship between the semantic content and the effect of relieving driver fatigue. The next involves comparing interesting auditory stimulation with existing effective fatigue-relieving methods, such as the use of electrodes for head-resting, to further improve these methods. Finally, we intend to use our method to alleviate driver fatigue in real driving situations.

## 5. Conclusions

The main contribution of this study is that the VMD-MMSE method was applied to the detection of driving fatigue and interesting auditory stimulation was used to alleviate driving fatigue. Furthermore, we found that the VMD-MMSE method was capable of detecting driving fatigue more effectively, compared with the SampEn method. In this paper, the cognitive load of the subjects was appropriately increased through exposure to interesting auditory stimulation to alleviate driving fatigue. These results indicate that interesting auditory stimulation can effectively relieve driving fatigue. In addition, the method does not affect normal driving, and is easy to use in normal driving.

## Figures and Tables

**Figure 1 entropy-23-01209-f001:**
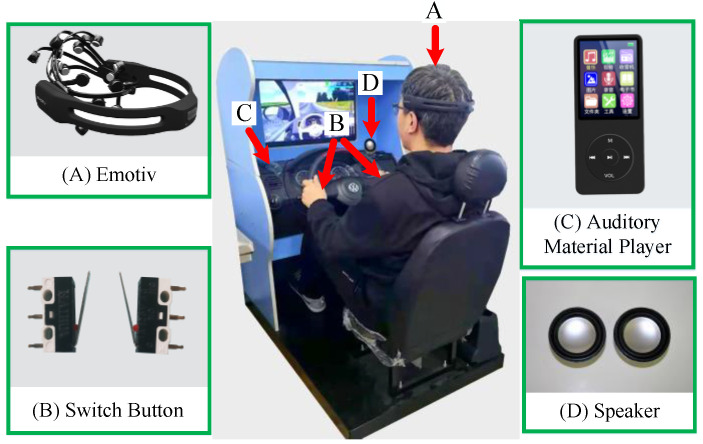
Experimental setup.

**Figure 2 entropy-23-01209-f002:**
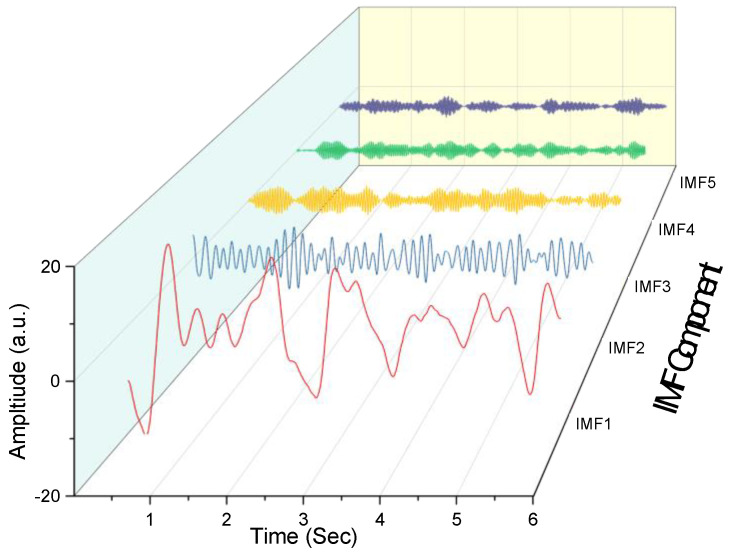
IMF component.

**Figure 3 entropy-23-01209-f003:**
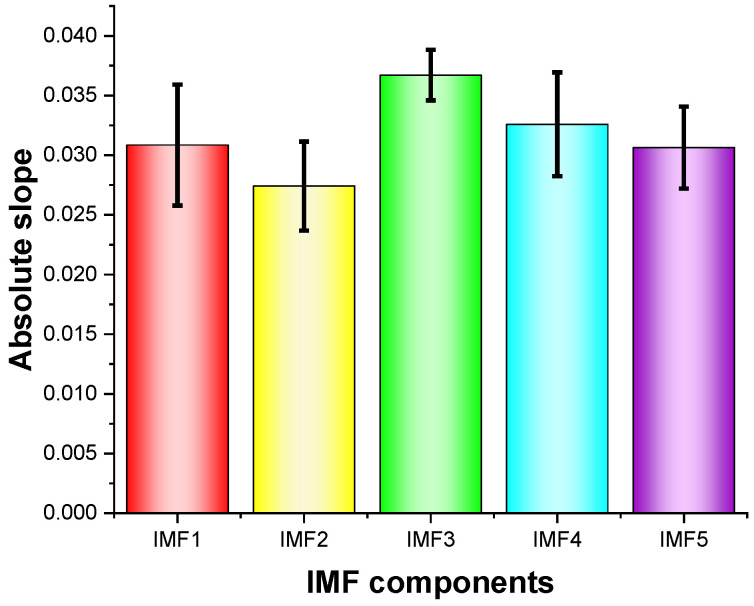
Single-scale entropy change trends/slopes (absolute values) for five IMFs.

**Figure 4 entropy-23-01209-f004:**
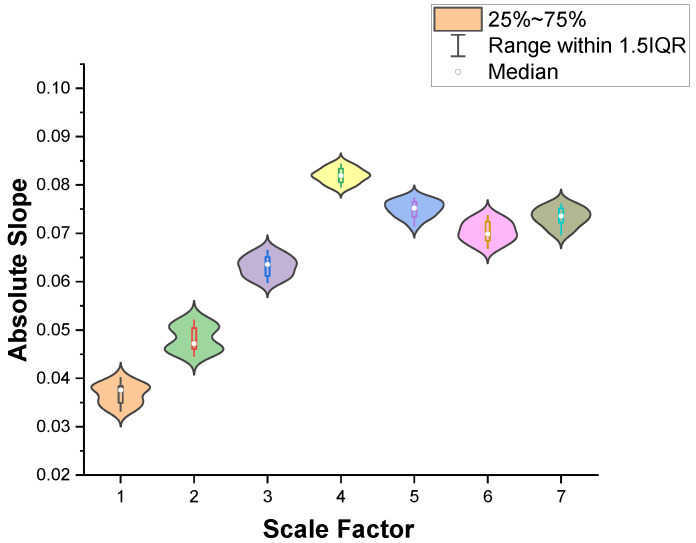
The relation between absolute slope and the scale factor.

**Figure 5 entropy-23-01209-f005:**
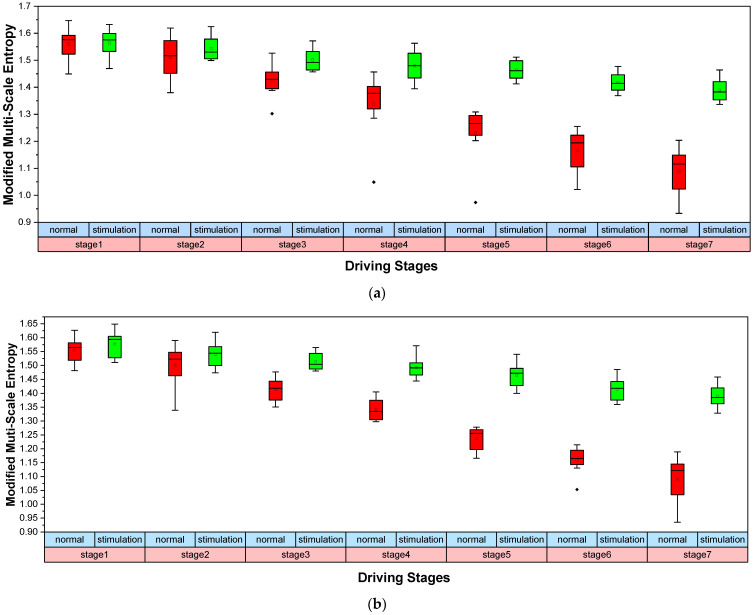
MMSE results of the two driving modes. (**a**) Variations in MMSE characteristics of the O1 channel in the two driving modes; (**b**) Variations in MMSE characteristics of the O2 channel in the two driving modes.

**Figure 6 entropy-23-01209-f006:**
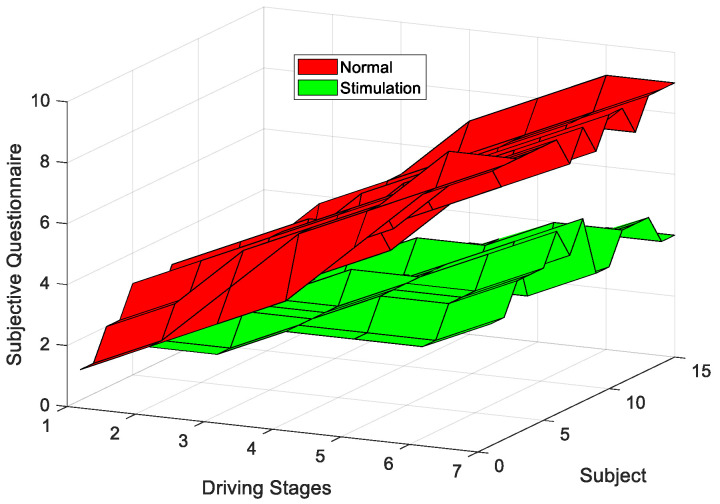
Changes in questionnaire scores over the driving stages.

**Figure 7 entropy-23-01209-f007:**
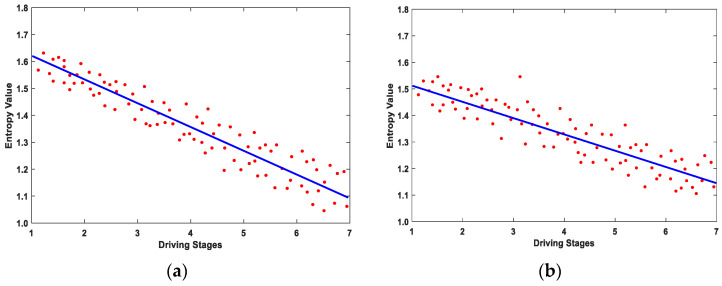
Comparison of the trends of VMD-MMSE and SampEn characteristics in a subject. (**a**) Fitting of VMD-MMSE feature changes using LSM; (**b**) Fitting of SampEn feature changes using LSM.

**Table 1 entropy-23-01209-t001:** Analysis results of linear-mixed effects model.

Source	Numerator Degrees of Freedom	Denominator Degrees of Freedom	F	Significance
intercept	1	100	16,573.398	6.32 × 10^−113^
gender	1	100	2.761	0.315
age	2	100	2.070	0.473
time	1	100	895.094	1.07 × 10^−51^

**Table 2 entropy-23-01209-t002:** Comparison of characteristic change trends.

	VMD-MMSE	SampEn
Subject1	−0.08301	−0.05894
Subject2	−0.08075	−0.06331
Subject3	−0.08229	−0.0576
Subject4	−0.07985	−0.05903
Subject5	−0.08089	−0.06074
Subject6	−0.08315	−0.05638
Subject7	−0.08161	−0.06208
Subject8	−0.08405	−0.06065
Subject9	−0.07863	−0.05447
Subject10	−0.08357	−0.05872
Subject11	−0.08304	−0.05679
Subject12	−0.08142	−0.05894
Subject13	−0.08059	−0.05923
Subject14	−0.08391	−0.05272
Subject15	−0.07584	−0.06054
mean	−0.08151	−0.05868
S.D. ^1^	0.00215	0.00266

^1^ S.D. denotes the standard deviation.

## Data Availability

The effective data link: https://pan.baidu.com/s/1YOb_1D9z51wLWmbWcNT5Ww (Extraction code: s7l9).

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
