# Peer review of "Research on Driving Fatigue Alleviation Using Interesting Auditory Stimulation Based on VMD-MMSE"

_entropy, 2021, doi:10.3390/e23091209_

Round 1
Reviewer 1 Report
This paper presents a study on driving fatigue detection and alleviation using interesting auditory stimulation based on modified multi-scale entropy (MMSE) with variational mode decomposition (VMD) algorithm.
The manuscript is well written and the reported results and findings are very interesting.
One minor comment: an exploration on AM-FM decomposition techniques, e.g. the empirical mode decomposition (EMD) or the iterated Hilbert transform (IHT), should be added in the introduction section.
Author Response
Dear reviewer,
Thank you for your comments and suggestions for my manuscript. I have modified the manuscript accordingly, and the detailed corrections are listed below point by point.
Comments and Suggestions for Authors
This paper presents a study on driving fatigue detection and alleviation using interesting auditory stimulation based on modified multi-scale entropy (MMSE) with variational mode decomposition (VMD) algorithm.
The manuscript is well written and the reported results and findings are very interesting.
One minor comment: an exploration on AM-FM decomposition techniques, e.g. the empirical mode decomposition (EMD) or the iterated Hilbert transform (IHT), should be added in the introduction section.
Thank you for your careful review for our manuscript. We have described the AM-FM decomposition techniques according to your comment, as shown below. Additionally, we have added it to the introduction section.
In recent years, non-stationary signal analysis has been paid more and more attention by researchers, in which FM-AM signal analysis is of great significance in non-stationary signal analysis. The frequency and amplitude information of non-stationary signals can be obtained by demodulate FM-AM signals. Qin achieved FM-AM decomposition of multi-components by combining energy decomposition and adaptive filtering [1]. At present, AM-FM decomposition techniques mainly include the iterated Hilbert transform (IHT) [2], the empirical mode decomposition (EMD) [3], the local mean decomposition (LMD) [4], etc. The IHT primarily separates the amplitude signal from the original signal, and then filter is used to separate the single component from the amplitude signal. Multiple FM-FM single component can be extracted from the original signal by iterative use of Hilbert transform and filter. EMD is an adaptive signal processing method that adaptively decomposes complex multi-component signals into the sum of several Intrinsic Mode functions (IMF). Then Hilbert transformation is performed on to calculate instantaneous frequency and instantaneous amplitude of each IMF component. In this study, EEG was adaptive decomposed using the VMD to obtain multiple IMF components, in which IMF was defined as FM-AM signal. The VMD can effectively separate noise signals through adaptive decomposition, which is more conducive to driving fatigue feature extraction.
[1] Qin, Y. Multicomponent AM-FM demodulation based on energy separation and adaptive filtering. Mechanical Systems and Signal Processing. 2013, 38(2), 440-459.
[2] Qin, Y., Qin, S. R., Mao, Y. F. Research on iterated Hilbert transform and its application in mechanical fault diagnosis. Mechanical Systems and Signal Processing. 2008, 22(8), 1967-1980.
[3] Munoz-Gutierrez, P. A., Giraldo, E., Bueno-Lopez, M., Molinas, M. Localization of Active Brain Sources From EEG Signals Using Empirical Mode Decomposition: A Comparative Study. Frontiers in Integrative Neuroscience. 2018, 12, 55.
[4] Smith, J. S. The local mean decomposition and its application to EEG perception data. Journal of the Royal Society Interface. 2005, 2(5), 443-454.

Reviewer 2 Report
The authors investigated driving fatigue by means of EEG quantitative analysis and they concluded that auditory stimulation can alleviate the cognitive load. However, there are serious flaws in the study that have to be addressed as soon as possible to consider the manuscript for publications. Ideas are not fully original and the lack of coherence in scientific presentation can hinder the study.
- Revise the overall English. You may want to use terms like “qualitative and quantitative approaches” instead of “subjective and objective methods”.You may also want to split the sentences, such as the text from “Considering” to “[23]” in the third paragraph of the introduction (too long).
- State your objective clearly. The last paragraph of the introduction does not say what the study is about.
- Explain the methods. A random selection of 8 subjects out of 15 is not a scientific procedure and you want to delve into the reason for the selection. You want also to describe the setup for both the experiment and the data analysis.
- Explain the filtering. What do you mean by “EEG LAB software was used to perform baseline correction of the raw EEG”? Which method did you use?
- The text quality of both VMD and Sample Entropy should be improved. On the contrary, do not explain OLS.
- The statistical analysis is pretty basic. What about ANOVA and multi comparisons?
- You must explain in the methods how you selected IMF3 and scale 4 by showing the pipeline. On top of that, it is not clear if the slopes are all significant for all subjects. Provide also more background about the occipital channels selection.
- Figure 7 should be done on all subjects and Table 1 should provide statistical background to support this statement.
Author Response
Dear reviewer,
Thank you for your comments and suggestions for my manuscript. I have modified the manuscript accordingly, and the detailed corrections are listed below point by point.
Comments and Suggestions for Authors
The authors investigated driving fatigue by means of EEG quantitative analysis and they concluded that auditory stimulation can alleviate the cognitive load. However, there are serious flaws in the study that have to be addressed as soon as possible to consider the manuscript for publications. Ideas are not fully original and the lack of coherence in scientific presentation can hinder the study.
- ①Revise the overall English. You may want to use terms like “qualitative and quantitative approaches” instead of “subjective and objective methods”.
Thank you for your careful review for our manuscript. In our study, the subjective method mainly refers to the questionnaire (such as the Karolinska Sleepiness Scale), which mainly relies on the subjective feelings of the subjects, such as extremely alert, alert, neither alert nor sleepy, some signs of sleepiness, sleepy, very sleepy etc. The researcher obtains the fatigue level of the subjects according to the scores of the subjects. We consider that the method is quantitative method. In addition, most researchers divide the analysis methods of fatigue driving research into "subjective method" and "objective method" [1-3].
[1] Fu, R. R., Wang, H., & Zhao, W. B. (2016). Dynamic driver fatigue detection using hidden Markov model in real driving condition. Expert Systems with Applications, 63, 397-411.
[2] Wang, F. W., Wang, H., & Fu, R. R. (2018). Real-Time ECG-Based Detection of Fatigue Driving Using Sample Entropy. Entropy, 20(3).
[3] Jiao, K., Li, Z. Y., Chen, M., Wang, C. T., & Qi, S. H. (2004). Effect of different vibration frequencies on heart rate variability and driving fatigue in healthy drivers. International Archives of Occupational and Environmental Health, 77(3), 205-212.
② You may also want to split the sentences, such as the text from “Considering” to “[23]” in the third paragraph of the introduction (too long).
Thank you for your careful review for our manuscript. We are ashamed to make these mistakes. We have made revisions to your comments, and the revision results are as follows:
1)It is not obvious that the mental fatigue characteristics in some leads, and the noise interference of redundant channels will increase the computational complexity. Li et al. identified the best indicator of driving fatigue by grey relational analysis, and reduced the dimension of effective electrode by kernel principal component analysis, and finally obtained two important leads (FP1 and O1)
2)After reviewing the whole manuscript, we found that the sentence “In order to select the IMF component that containing the most obvious driving fatigue characteristics from the five IMF components, we used the LSM to perform one linear fit to the single-scale entropy change trends of the five IMFs in the seven driving stages and calculated the slope of the fatigue change trends in the seven driving stages.” also had similar problem, and we also made corresponding revisions. The revised sentence is “Each IMF component contains different fatigue components, therefore, it is necessary to select the IMF component with the most obvious driving fatigue characteristics from the five IMF components. In this study, we used the LSM to perform one linear fit to the single-scale entropy change trends of the five IMFs in the seven driving stages and calculated the slope of the fatigue change trends in the seven driving stages.”
- State your objective clearly. The last paragraph of the introduction does not say what the study is about.
Thank you for your careful review for our manuscript. We have made revisions to your comments, adding a paragraph at the end of the original introduction to clearly state the purpose of the study, as detailed follow:
In this study, the VMD algorithm was firstly used to adaptive decompose drivers’ EEG signals, obtained multiple IMF. Secondly, the multi-scale entropy features of each IMF were calculated by multi-scale sample entropy algorithm. Finally, multi-scale sample entropy was used to detect driving fatigue. Due to driving fatigue alleviation is of great significance to traffic safety. Therefore, driving fatigue alleviation was studied from the perspective of driver cognitive load in this paper, that is, using the method of interesting auditory stimulation to make the driver’s cognitive level reached relatively best state, achieved the purpose of driving fatigue alleviation.
- ① Explain the methods. A random selection of 8 subjects out of 15 is not a scientific procedure and you want to delve into the reason for the selection.
Thank you very much for your valuable review comment. In the initial experiment, we randomly numbered 15 subjects from 1 to 15. During the experiment, it was considered that the subjects needed to wait 15 days after finishing the first type of experiment before performing the second type of experiment. During these 15 days, the subjects may experience unexpected conditions such as illness, which may affect the results of the experiment. Therefore, we selected the first eight subjects for the experiment, then did the same experiment for the remaining seven subjects. In addition, we believe that the data of 8 subjects can support the conclusion. After taking your comment into account and referring to other literature, we all agree that it is really too few to choose 8 subjects to participate in the experiment. Consequently, the data of the original 15 subjects were re-analyzed by adding the data of the other 7 subjects to the whole experiment.
②You want also to describe the setup for both the experiment and the data analysis.
Thank you very much for your valuable review comment. We have added the following contents to the “Methods” section. In addition, the experimental procedures have been supplemented, please see the section “Subjects”.
- ① Explain the filtering. What do you meanby “EEG LAB software was used to perform baseline correction of the raw EEG”?
Thank you very much for your valuable review comment. In this study, EEGLAB software was used to pre-process EEG signals. EEGLAB Software is a toolbox for MATLAB, which can perform independent component analysis (ICA), Time/Frequency analysis, baseline correction, artifact rejection, etc. In our study, we use the baseline correction function of EEG LAB software to achieve baseline correction of raw EEG signals.
②Which method did you use?
Thank you very much for your valuable review comment. In this study, we used the median filter correction method in EEGLAB software for baseline correction. Median filter is a nonlinear low-pass filter. The basic principle of median filter is to replace a certain point in the sequence with the median value of a local area to eliminate isolated noise points. The median filter correction method is to subtract the output of the low-pass filter from the original signal to eliminate baseline drift.
- The text quality of both VMD and Sample Entropy should be improved. On the contrary, do not explain OLS.
This is a constructive suggestion for our manuscript. We have made corresponding revisions as you suggested. We provide more detailed introduction to the VMD and SampEn methods (as shown in “Methods” section). In addition, we simplify introduction of the LSM. The revised content is shown in “Methods” section.
- The statistical analysis is pretty basic. What about ANOVA and multi comparisons?
Thank you for your careful review for our manuscript. In this study, the results of normal driving mode and interesting auditory stimulation mode were analyzed to demonstrate the effectiveness of interesting auditory stimulation for alleviating driving fatigue. Consequently, this study was based on two types of data, which could not perform multi comparisons analysis. We referred to you comment, used one-way ANOVA to analyze the fatigue characteristics calculated by MMSE for two types of experiments. The results for O1 (F=4.78>F(1,12)=4.75, p=0.0493<0.05) and O2 (F=4.98>F(1,12)=4.75, p=0.0455<0.05). In addition, we conducted one-way ANOVA on the KSS scores for the two types of experiments, the result for F=4.87>F(1,12)=4.75, p=0.0475<0.05.
- 7. ① You must explain in the methods how you selected IMF3 and scale 4 by showing the pipeline. On top of that, it is not clear if the slopes are all significant for all subjects.
Thank you for your careful review for our manuscript. In our manuscript, the least square method was used to fit the change trend of fatigue characteristics of 15 subjects during the 3 hour driving task. If the change trend of extracted fatigue features is obvious, it is beneficial to the detection of driving fatigue, otherwise, it is not conducive to the detection of driving fatigue. As can be seen from the bar graph in Figure 3, the absolute value of the slope mean fitted by the change trend of the single-scale entropy feature of IMF3 component is the largest, which is more conducive to the detection of driving fatigue. Moreover, the standard deviation of IMF3 component is the smallest, indicating that the fatigue characteristics change trend of IMF3 component is stable, which conducive to driving fatigue detection. Therefore, IMF3 component was selected to detect driving fatigue in the paper. It can be seen from the violin plot in Figure 4 that the absolute slope of fatigue characteristics change trend corresponding to scale 4 is the largest, and the distributed of the absolute slope of fatigue characteristics change trend corresponding to scale 4 is more centralized, indicating that scale 4 has less impact on individual differences, which is beneficial to detect driving fatigue. Consequently, the multi-scale entropy algorithm selects the scale as 4.
② Provide also more background about the occipital channels selection.
Thank you for your careful review for our manuscript. We have added the more background (please see the next paragraph for details) about the occipital channels selection and added it to the introduction section.
Jap et al. calculated the coherence between the inter-hemispheres of θ, δ, α and β sub-bands of five homologous EEG electrode pairs (FP1-FP2, C3-C4, T7-T8, P7-P8, and O1-O2). The results showed that the inter-hemispheric coherence values of the frontal and occipital were higher than the central, parietal and temporal in the four frequency bands [1]. Wang et al. collected EEG signals from the central, parietal and occipital regions of the subjects, and used G-P algorithm to evaluate the correlation dimension to quantify EEG signals. The results showed that the correlation dimension of each channel decreased significantly when subjects were in fatigue state [2].
[1] Jap, B. T., Lal, S., Fischer, P. Inter-hemispheric electroencephalography coherence analysis: assessing brain activity during monotonous driving. Int J Psychophysiol. 2010, 76(3), 169-173.
[2] Wang, J., Wu, Y.Y., Qu, H., Xu, G. EEG-based fatigue driving detection using correlation dimension. JOURNAL OF VIBROENGINEERING. 2014, 16(1), 407-413.
- 8. Figure 7 should be done on all subjects and Table 1 should provide statistical background to support this statement.
Thank you for your careful review for our manuscript. We have added data for all subjects to Figure 7, which is shown in the discussion section. In addition, we added the slope that fatigue characteristics change of the seven driving stages for 15 subjects calculated by the SampEn and VMD-MMSE methods to Table1, which is shown in the discussion section.
Table 1. Comparison of characteristic change trends
|
|
VMD-MMSE |
SampEn |
|
Subject1 |
-0.08301 |
-0.05894 |
|
Subject2 |
-0.08075 |
-0.06331 |
|
Subject3 |
-0.08229 |
-0.0576 |
|
Subject4 |
-0.07985 |
-0.05903 |
|
Subject5 |
-0.08089 |
-0.06074 |
|
Subject6 |
-0.08315 |
-0.05638 |
|
Subject7 |
-0.08161 |
-0.06208 |
|
Subject8 |
-0.08405 |
-0.06065 |
|
Subject9 |
-0.07863 |
-0.05447 |
|
Subject10 |
-0.08357 |
-0.05872 |
|
Subject11 |
-0.08304 |
-0.05679 |
|
Subject12 |
-0.08142 |
-0.05894 |
|
Subject13 |
-0.08059 |
-0.05923 |
|
Subject14 |
-0.08391 |
-0.05272 |
|
Subject15 |
-0.07584 |
-0.06054 |
|
mean |
-0.08151 |
-0.05868 |
|
S.D. |
0.00215 |
0.002688 |

Round 2
Reviewer 2 Report
I would like to thank the authors for the improved manuscripts. I still have a couple of suggestions
- Please improve your abstract and make sentences like the following one more professional and scientific: In addition, the method is simple to operate and easier to be ap- plied. in the future. How easier? Based on which metric?
- Could you discuss a comparison between your method and mutlifractality as reported in the two following papers? What about the difference between MSE with laboratory data and your experiment? Please refer at the following two papers.
- https://www.frontiersin.org/articles/10.3389/fphys.2020.00741/full
- https://www.frontiersin.org/articles/10.3389/fphys.2020.581250/full
- Referring to the second link, could you compute a linear-mixed effect model for figure 7? Or can you discuss the limits of your regression in the dedicated section?
Author Response
Dear reviewer,
Thank you for your comments and suggestions for my manuscript. I have modified the manuscript accordingly, and the detailed corrections are listed below point by point:
Comments and Suggestions for Authors
I would like to thank the authors for the improved manuscripts. I still have a couple of suggestions
- Please improve your abstract and make sentences like the following one more professional and scientific: In addition, the method is simple to operate and easier to be applied in the future. How easier? Based on which metric?
Thank you for your careful review for our manuscript. We have made revisions to your comments, adding it to the ‘Abstract’ section. The revision results are as follows:
In addition, the interesting auditory stimulation method, which only needs to play interesting auditory information on the vehicle-mounted player, can effectively relieve driving fatigue. Compared with traditional driving fatigue relieving methods, such as sleeping and drinking coffee, this interesting auditory stimulation method can relieve fatigue in real-time when the driver is driving normally.
- Could you discuss a comparison between your method and mutlifractality as reported in the two following papers? What about the difference between MSE with laboratory data and your experiment? Please refer at the following two papers.
- https://www.frontiersin.org/articles/10.3389/fphys.2020.00741/full
- https://www.frontiersin.org/articles/10.3389/fphys.2020.581250/full
Thank you for your careful review for our manuscript. According to your comment, we compared mutlifractality with MMSE, adding it to the ‘Introduction’ section.
EEG signals are highly complex signals of non-linear and non-stationary, therefore, the linear analysis methods cannot well reflect the internality dynamic characteristics of EEG signals. As one of the non-linear analysis methods, mutlifractality method can reveal the regularity problems within the non-linear system by studying its dynamic characteristics such as disorder, irregularity and uncertainty. At present, the mutlifractality method is widely used in physiological signal processing [1-3]. Additionally, as one of the non-linear processing method, entropy is also widely used in driving fatigue detection [4-6]. Compared with mutlifractality method [7,8], MMSE can extract EEG features on time scale, and the data length required by the MMSE method is shorter in the EEG processing. Consequently, for special situation of driving fatigue detection, the MMSE method can better meet the needs of traffic safety.
[1] Lavanga, M., Bollen, B., Jansen, K., et al. A Bradycardia-Based Stress Calculator for the Neonatal Intensive Care Unit: A Multisystem Approach. Frontiers in Physiology. 2020, 11, 741.
[2] Lavanga, M., Heremans, E., Moeyersons, J., et al. Maturation of the Autonomic Nervous System in Premature Infants: Estimating Development Based on Heart-Rate Variability Analysis. Frontiers in Physiology. 2021, 11, 581250.
[3] Lavanga, M., De Ridder, J., Kotulska, K., et al. Consortium, E. Results of quantitative EEG analysis are associated with autism spectrum disorder and development abnormalities in infants with tuberous sclerosis complex. Biomedical Signal Processing and Control. 2021, 68, 102658.
[4] Kar, S., Bhagat, M., Routray, A. EEG signal analysis for the assessment and quantification of driver's fatigue. Transportation Research Part F-Traffic Psychology and Behaviour. 2010, 13(5), 297-306.
[5] Mu, Z.D., Hu, J.F., Min, J.L. Driver Fatigue Detection System Using Electroencephalography Signals Based on Combined Entropy Features. Applied Sciences-Basel. 2017, 7(2), 150.
[6] Wang, F., Wu, S.C., Ping, J.Y., et al. EEG Driving Fatigue Detection With PDC-Based Brain Functional Network. IEEE Sen-sors Journal. 2021, 21(9), 10811-10823.
[7] Li, G., Duan, S., Yu, H., Guan, D. Study on characteristic parameters of engine vibration signal based on multi-fractal. Transactions of CSICE. 2008, 26(1), 87-91.
[8] Zhang, T., Wang, H., Chen, J., He, E. Detecting Unfavorable Driving States in Electroencephalography Based on a PCA Sample Entropy Feature and Multiple Classification Algorithms. Entropy. 2020, 22(11), 1248.
- Referring to the second link, could you compute a linear-mixed effect model for figure 7? Or can you discuss the limits of your regression in the dedicated section?
Thank you very much for your valuable review comment. According to your comment, we compute a linear-mixed effect model for figure 7, adding it to the ‘Discussion’ section. The analysis results are shown in the following Table1.
Studies have shown that these three factors: drivers’ age, gender and driving time, are the main factors affecting drivers’ driving fatigue [1,2]. In this study, the effects of drivers’ age, gender and driving time on driving fatigue were analyzed using linear-mixed effect model. The results of our analysis using SPSS software are shown in Table 1.
Table 1. Analysis results of linear-mixed effect model.
|
Source |
numerator degree of freedom |
denominator degree of freedom |
F |
significance |
|
intercept |
1 |
100 |
16573.398 |
6.32×10-113 |
|
gender |
1 |
100 |
2.761 |
0.315 |
|
age |
2 |
100 |
2.070 |
0.473 |
|
time |
1 |
100 |
895.094 |
1.07×10-51 |
As can be seen from Table 1, the drivers’ age and gender have no significant influence on the degree of driving fatigue (P>0.05), however, the driving time has significant influence on the degree of driving fatigue (P<0.05). Therefore, the LSM was used to linear fit the driving time and entropy characteristics of subjects in this study.
[1] Chai Meng. Identification and Early-warning of Fatigue State of Intercity Coach Drivers. Thesis for Doctor’s degree, Jilin University, Changchun City, 2019.
[2] Jin Jian. Evaluation of Driving Fatigue Mechanism and Feedback Selecting Model. Thesis for Doctor’s degree, Southwest Jiaotong University, Chendu City, 2002.
